# Prevalence of Stress Urinary Incontinence and Risk Factors among Saudi Females

**DOI:** 10.3390/medicina59050940

**Published:** 2023-05-13

**Authors:** Abdulrahim M. Gari, Ethar H. Alhashmi Alamer, Rania O. Almalayo, Wafa A. Alshaddadi, Sadin A. Alamri, Razan S. Aloufi, Saeed Baradwan

**Affiliations:** 1Department of Obstetrics and Gynecology, Faculty of Medicine, Umm Al-Qura University, Makkah 24382, Saudi Arabia; 2Department of Obstetrics and Gynecology, King Faisal Specialist Hospital and Research Center, Jeddah 11211, Saudi Arabia; 3College of Medicine, Umm Al-Qura University, Makkah 24382, Saudi Arabia; 4College of Medicine, University of Jeddah, Jeddah 21959, Saudi Arabia; 5College of Medicine, AlRayan Colleges, Al-Madinah 42541, Saudi Arabia

**Keywords:** prevalence, risk factors, Saudi female, stress urinary incontinence, stress incontinence

## Abstract

*Background and Objectives*: Stress urinary incontinence (SUI) is involuntary urine leakage upon effort or physical exertion, sneezing, or coughing, and it is the most prevalent type of urinary incontinence (UI) in women. We aimed to estimate the prevalence of SUI and its risk factors among Saudi females. *Materials and Methods*: A descriptive cross-sectional study was conducted in the Kingdom of Saudi Arabia between March 2022 and July 2022, with a total of 842 respondents. We included Saudi females over the age of 20 years. Data were collected through an online questionnaire distributed to the target group and analyzed using SPSS software. *Results*: The prevalence of SUI was found to be 3.3% among Saudi women. Moreover, only 41.8% of the participants had at least one pregnancy; the majority had five or more pregnancies (29%). According to our findings, the majority of the participants diagnosed with SUI had the following risk factors: increased age, widowhood, a family history of SUI, and a history of pregnancy. The results revealed that the odds of SUI increased among Saudi females with a family history of SUI by 19.68-fold compared with those who had no family history of SUI, and this was statistically significant (*p* < 0.001). *Conclusion*: The prevalence of SUI among Saudi females was found to be relatively low. The above-listed associated factors should be considered in future research and interventions.

## 1. Introduction

The International Urogynecological Association (IUGA) describes urinary incontinence (UI) as the complaint of involuntary loss (leakage) of urine [1]. A subtype of UI is stress urinary incontinence (SUI), which is characterized by weakness of the pelvic floor muscles that allows leakage of urine when abdominal pressure is increased due to behaviors such as laughing, sneezing, coughing, and climbing stairs [2]. Many factors affect the strength of the urinary bladder, which can lead to the development of SUI. Previous studies have shown that SUI has a higher prevalence in smokers, either due to smoking itself or because of smoking-related illnesses that lead to chronic coughing [3]. It also occurs frequently in people with an elevated body mass index (BMI) [4]. SUI is also affected by one’s diet; the long-term dietary intake of starchy foods with a low intake of high-fiber foods may be a risk factor for constipation, which increases abdominal pressure and causes SUI [5,6]. The prevalence of SUI is significantly higher in females who undergo vaginal delivery compared with those who undergo cesarean section, and the prevalence also depends on the number of previous vaginal deliveries. Vaginal delivery is associated with an 8% increase in the risk of SUI compared with delivery by cesarean section [7]. Dystocia has been shown to be one of the risk factors for SUI [8]. Furthermore, females who have reached menopause or use oral contraceptives are at a higher risk of developing SUI due to decreased estrogen levels [9]. Additionally, a comprehensive review has highlighted strong evidence for increased rates of SUI among women who are physically active or play strenuous sports [10]. In general, UI, regardless of its type, has both physical and psychological implications, as well as being associated with a lower quality of life, and it imposes a considerable societal and financial burden [11]. Previous research has found variations in the prevalence of SUI. The rate of SUI among Saudi females was found to be 36.4% in one study [12], and 50% in another study [13]. However, there is no information on the present prevalence and recent changes in UI and its subtypes among women in the Kingdom of Saudi Arabia (KSA). The objective of our study was to determine the prevalence of SUI and its risk factors among Saudi females.

## 2. Materials and Methods

A descriptive cross-sectional study was conducted in the KSA between March 2022 and July 2022. Ethical approval was obtained from the Institutional Review Board (IRB) at Umm Al-Qura University, Makkah, Saudi Arabia (Letter No. HAPO-02-K-012-2022-03-993, dated 3 March 2022). All participants provided informed written consent. This study was conducted according to the ethical standards given by the Declaration of Helsinki.

We included all Saudi females over the age of 20 years. Any females younger than 20 years of age were excluded, as were all non-Saudi females and Saudi females with endocrine disease, dementia, delirium, neurodegenerative changes, and osteodegenerative changes. Survey responses were collected anonymously. The sample size required for this study was calculated using OpenEpi version 3.0. With an estimated population size of approximately 15 million, a 95% confidence interval (CI) and a 5% margin of error, a sample size of 385 participants was considered in this study. However, to avoid any potential exclusions, we aimed to collect more data than the calculated sample size.

A self-administered questionnaire was designed using Google Forms. The questionnaire was developed based on previous studies [12,13,14], modified for the targeted population and reviewed by three expert obstetricians and gynecologists to ensure clarity and simplicity. It was available in the Arabic language for increased ease of use for the targeted population. The developed questionnaire contained five main sections: socio-demographic data, number of cases of SUI, risk factors, the International Consultation on Incontinence Questionnaire Urinary Incontinence Short Form (ICIQ-UI SF), and the consent form on the first page. The questionnaire was distributed electronically to the targeted group, after receiving approval by the IRB, through different social media apps, including WhatsApp, Twitter, and Telegram. Data were collected online via Google Forms and were inputted and stored in a widely available spreadsheet format (Microsoft Excel). 

Statistical software (IBM SPSS version 23) was used, and descriptive statistics were gathered to summarize the data and synthesize and report the variables. Proportions, frequencies, means, and standard deviations for continuous variables were also included in the data description when appropriate. Student’s t-test was used to compare numerical variables, whereas the chi-square test was used to compare categorical variables. Logistic regression analysis using odds ratio (OR) and a 95% CI was used to assess the significance of several socio-demographic predictors (i.e., age, gender, nationality, marital status, educational level, occupational status, monthly income, presence of reported diagnosis of SUI, family history of SUI, previous pregnancy, number of pregnancies, number of normal deliveries, number of cesarean sections, and smoking status) in terms of the likelihood of developing SUI. The OR was calculated by dividing the odds of the one group (experimental) by the odds in the second group (control). For all purposes, a p-value of less than 0.05 established statistical significance.

## 3. Results

Out of 1229 questionnaires, 1221 participants agreed to participate in the study, representing a response rate of 99.3%. After applying the exclusion criteria (64 males, 66 non-Saudi, 110 aged <20 years, and 139 with diseases such as endocrine disease and dementia), 842 participants were included in the study. The most common age group was 20–29 years (57%). Approximately one-third of the study population (33.4%) resided in the central region of the KSA, followed by the western (30.8%), eastern (14.7%), and southern regions (10.9%). More than half of the participants were single (50.4%). Regarding the participants’ educational level, 73.3% had a bachelor’s degree or higher. A minority of the participants reported having no monthly income (26.8%). The mean weight of participants was found to be 65.5 ± 17.1 kg, whereas the mean height was 157.6 ± 15.6 cm. Regarding BMI, most participants fell in the healthy range (43.2%). Further details regarding the socio-demographic characteristics of the participants are shown in Table 1. 

We found the rate of SUI to be 3.3% (n = 28), whereas 96.7% of participants did not have SUI (n = 814) (Figure 1). The majority of the participants denied a family history of SUI (64.6%). Among those who reported having SUI, the majority reported experiencing it once per week (15.7%), with 21.7% reporting the leakage volume as minimal and as occurring mostly during laughing, exercising, sneezing, or heavy lifting (18.1%). Only 41.8% of the participants had a history of pregnancy, and 29% of our participants reported having five or more pregnancies. Among women who had been pregnant before, 20.2% reported that five or more of their pregnancies were delivered vaginally, and 20.2% had at least one cesarean section. Only 4.2% of the participants reported smoking. Half of the participants who smoked had never suffered from a chronic cough due to smoking (51.4%). Further details are shown in Table 2.

We found that the presence of a reported diagnosis of SUI increased with age (*p* < 0.05). The majority of those diagnosed with SUI were widowed (16.7%). The majority of the participants diagnosed with SUI had a monthly income of more than 8,000 SR (n = 19, 9.6%), were obese (n = 19, 10.4%), and had a family history of SUI. Among the SUI-diagnosed participants, only 7.1% had been pregnant before and had a higher BMI. Ultimately, age, marital status, family history of SUI, history of pregnancy, and BMI were found to be significantly associated with SUI (*p* < 0.05). Further details are highlighted in Table 3.

Regarding the impact of SUI on respondents’ everyday life, the vast majority of participants reported a slight impact on their life (81.8%); however, 69.1% of participants demonstrated an impact score of zero. The mean impact score of SUI on the everyday life of participants was 1.28 ± 2.7 out of 10. Most of the participants with an impact score of 5 or greater (n = 99) had never been diagnosed with SUI before (n = 73, 73.7%). The mean ICIQ-UI SF score was found to be 2.4 ± 4.2 out of 21. Additionally, our results found that only 4.3% and 0.6% of our participants revealed a severe or very severe impact of SUI on their quality of life, respectively. Table 4 depicts a summary of the impact of SUI on participants’ everyday life.

There was a significant association between age and ICIQ-UI SF score (*p* < 0.001); the effect increased with age, as depicted by the highest percentage of a slight effect of SUI on quality of life reported by participants in the 20–29-year age group, while the highest percentage of a severe effect on quality of life was observed in participants aged 60 years or more. There was also a significant association between BMI, the presence of a reported diagnosis of SUI, and ICIQ-UI SF score (*p* < 0.001); the highest percentage of a very severe effect of SUI on quality of life was observed in obese participants and participants with a reported diagnosis of SUI (Table 5).

The results revealed that the odds of SUI increased significantly among Saudi females with a family history of SUI by 19.68-fold when compared with those who had no family history of SUI (*p* < 0.001). However, smoking increased the odds of SUI by nine-fold when compared with non-smokers, but this result was not statistically significant (*p* = 0.052). Other variables did not have a significant effect on the odds of SUI among Saudi females. Further information is provided in Table 6.

## 4. Discussion

This study aimed to investigate the prevalence of SUI and its risk factors among Saudi women. We found that the prevalence of SUI was only 3.3%, which is lower than the figures previously reported in Ethiopia, Nigeria, and Turkey (11.4%, 21.1%, and 28.1%, respectively) [15,16,17]. The reason for this discrepancy remains unclear, but it may be attributed to the poor antenatal care follow-up in these countries compared with the KSA, as concluded by Alanazy et al. [18] Additionally, most of the participants were <30 years of age and had a lower likelihood of developing SUI.

In our study, we found that the presence of SUI increased with the number of pregnancies, and among women who had at least one vaginal delivery and no delivery via cesarean section. The likely explanation for this is that mechanical strain during repetitive delivery may cause muscle, fascia, and ligamentous disruption, as well as damage to connective and neurological structures of the pelvic organs and pelvic floor. Similar findings have been reported in two congruent studies The first study was conducted by Gyhagen et al., which reported up to a 20–30% increase in the prevalence of SUI due to pregnancy, and this increased to 43% if the delivery was vaginal [19]. The second study carried out by Altman et al. reported an increase in the prevalence of SUI by 12% due to pregnancy and vaginal delivery [20].

Older age was found to be significantly associated with a higher prevalence of SUI (*p* < 0.05). This may be explained by the fact that an increase in age is accompanied by weakness in the pelvic muscles and ligaments. This result was also found in other studies, such as that by Okunola et al. [16]. Unlike the results of the present study, Ozerdo et al. and Abdullah et al. [21,22] reported no significant correlation between SUI and age. However, the protective effect of cesarean delivery was consistent throughout the literature [23,24]. In the present study, a cesarean section was not found to have a significant effect on the diagnosis of SUI. Adaji et al. reported that SUI was more common in those who had spontaneous vaginal deliveries in the past [25]. In their studies, Oliveira et al. reported that the risk of SUI was 2.5 times greater in women who delivered vaginally [23]. Our study discovered that having a diagnosis of SUI was associated with higher BMI (*p* < 0.05). Furthermore, we discovered a significant association between increased BMI and the presence of a reported diagnosis of SUI (*p* < 0.001). Similarly, many epidemiological studies indicated that increased BMI was a risk factor for SUI in pregnancy [26]. In contrast, Scarpa et al. found no such association when they considered a BMI ≥ 30 in the third trimester of pregnancy [27].

We discovered that there was a significant association between age, BMI, and ICIQ-UI SF score (*p* < 0.001); the highest percentage of severe and very severe effects of SUI on participants’’ quality of life was observed in older and obese participants compared with others. This was found to be consistent with the study conducted by Narçiçeği et al., who found that there was a significantly positive correlation between age, BMI, and ICIQ-UI SF score [28].

Another interesting finding was that smoking increased the risk of SUI by 9-fold compared with non-smoking. This finding, although not significant here, has been demonstrated previously in the literature by Adaji et al. [25]. Cigarette smoke contains carbon monoxide, which affects oxygen flow to physiological tissues and causes muscular atrophy and weakening, affecting the pelvic floor muscles. Smoking can cause chronic and/or frequent coughing, which raises bladder pressure and places significant strain on the pelvic floor muscles, potentially damaging the innervating nerves and exacerbating SUI. Other than carbon monoxide, nicotine also stimulates the detrusor muscle [29].

Family history was also reported in our study as a factor that increases the odds of SUI by 19 times (*p* < 0.05). This finding may be related, in part, to genetic factors affecting collagen and other connective tissues responsible for the strengthening of body ligaments and muscles, including the pelvic floor muscles. This was found to be consistent with a study conducted by Ertunk et al., in which family history was found to be associated with participants developing UI more frequently than those with no family history of UI [30].

In our study, we found that most participants who reported a greater impact of SUI on their everyday life (i.e., a score of ≥5 out of 10) had never been diagnosed with SUI before (73.7%). In addition, 66.7% of the participants reported a severe impact of SUI on their quality of life, and 20% of participants who had reported a very severe impact had never been diagnosed with SUI before. Although SUI can affect quality of life, most women in our study had not previously been diagnosed with SUI, and this might be attributed to the shame and embarrassment of having SUI. This was found to be consistent with a study by Elenskaia et al., [31] who concluded that despite the high prevalence of SUI, it is still considered taboo, and the level of shame and embarrassment surrounding SUI is significantly higher than the feelings of shame and embarrassment that are associated with depression and cancer.

This study highlighted the prevalence of SUI among Saudi females. However, this study has limitations. The generalizability of the results is dubious, as this study lacked randomization, and the majority of the participants were under the age of 30 years old, which could have negatively impacted the true prevalence of SUI among the surveyed participants. The data were self-reported by participants. Additionally, the study did not examine key covariates, such as pelvic floor training and postpartum rehabilitation programs, which could have indirectly influenced the reported results. Detection and diagnosis of SUI via an online questionnaire may be suitable for epidemiologic or other informative research. However, the collection of data from outpatient clinics where relevant data could be obtained would have been more appropriate. Thus, future research needs to be conducted as a multi-center clinic-based survey or assessment with a large sample size and optimum regional coverage to estimate the true prevalence of SUI and its associated risk factors in the KSA.

## 5. Conclusions

The prevalence of SUI in this study was found to be lower (3.3%) than that reported in other studies. The results of this study may be useful to increase awareness in women during the pregnancy period by providing health training and consultancy services regarding SUI prevention and treatment.

Regular health education programs and interventions should be conducted to decrease the modifiable risk factors and increase women’s awareness of SUI management and associated factors. Pelvic training exercises are one of the most easily implemented interventions for treating and reducing the risk of SUI; thus, this should be available at the level of primary healthcare.

## Figures and Tables

**Figure 1 medicina-59-00940-f001:**
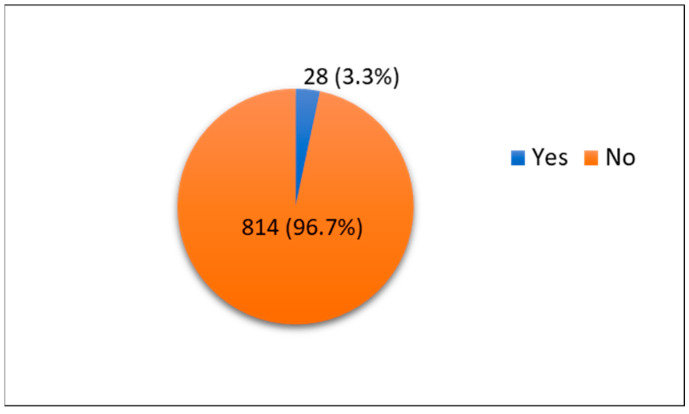
Prevalence of stress urinary incontinence among Saudi females investigated for urinary incontinence.

**Table 1 medicina-59-00940-t001:** Socio-demographic characteristics of the participants (n = 842).

Variable	Categories	Frequency	Percentage
Age groups (years)	20–29	480	57%
	30–39	190	22.6%
	40–49	112	13.3%
	50–59	53	6.3%
	60 or more	7	0.8%
Residence	Northern	86	10.2%
	Southern	92	10.9%
	Central	281	33.4%
	Eastern	124	14.7%
	Western	259	30.8%
Educational level	Primary	3	0.4%
	Intermediate	5	0.6%
	Secondary	153	18.2%
	Diploma	64	7.6%
	Bachelor’s degree or above	617	73.3%
Marital status	Single	424	50.4%
	Married	373	44.3%
	Divorced	39	4.6%
	Widowed	6	0.7%
Average monthly income (SAR)	Less than 4000	134	15.9%
	4000–8000	104	12.4%
	More than 8000	198	23.5%
	University incentive	180	21.4%
	No monthly income	226	26.8%
Body mass index (BMI) (kg/m^2^)	Underweight (<18.5)	58	6.9%
	Healthy weight (18.5–24.9)	364	43.2%
	Overweight (25–29.9)	237	28.1%
	Obese (>30)	183	21.7%
		Mean	SD ^1^
Weight (kg)		65.5	17.1
Height (cm)		157	15.6

^1^ SD: standard deviation, SAR: Saudi Riyal, kg/m^2^: kilogram per square meter, kg: kilogram, cm: centimeter.

**Table 2 medicina-59-00940-t002:** Associated factors of stress urinary incontinence among Saudi females investigated for stress urinary incontinence.

Variable	Categories	Frequency	Percentage
Previous conditions	Hypertension	6	0.7%
	Asthma	3	0.4%
	Depression	1	0.1%
	Uterine problems	2	0.2%
	Arthritis	2	0.2%
	Chronic constipation	1	0.1%
	PCOS ^1^	4	0.5%
	Rheumatism	2	0.2%
	Liver problems	1	0.1%
	None	827	98.2%
Family history of SUI ^2^	Yes	62	7.4%
	No	544	64.6%
	I do not know	236	28%
Pregnancy	Yes	352	41.8%
	No	490	58.2%
Number of pregnancies (n = 352)	1	61	17.3%
	2	65	18.5%
	3	52	14.8%
	4	72	20.5%
	5 or more	102	29%
Number of vaginal deliveries (n = 352)	0	71	20.2%
	1	67	19%
	2	56	15.9%
	3	44	12.5%
	4	43	12.2%
	5 or more	71	20.2%
Number of cesarean sections (n = 352)	0	212	60.2%
	1	71	20.2%
	2	31	8.8%
	3	20	5.7%
	4	10	2.8%
	5 or more	8	2.3%
Smoking status	Yes	35	4.2%
	No	807	95.8%
Number of cigarettes per day (n = 35)	10 or less	31	88.6%
	11–20	4	11.4%
	21–30	0	0%
Chronic cough (n = 35)	Never	18	51.4%
	Sometimes	12	34.3%
	Usually	2	5.7%
	Always	3	8.6%
Frequency of urinary incontinence	Never happens	621	73.8%
	Once per week or less	132	15.7%
	2–3 times per week	33	3.9%
	Once everyday	29	3.4%
	Multiple times everyday	22	2.6%
	All the time	5	0.6%
Amount of urine	Nothing	617	73.3%
	Small amount	183	21.7%
	Average amount	36	4.3%
	Large amount	6	0.7%
the presence of urinary incontinence	I don’t have urinary incontinence	594	70.5%
	Before reaching the toilet	101	12%
	During sneezing, coughing, or laughing	152	18.1%
	With active physical movements such as exercise or when lifting heavy objects	43	5.1%
	After using the toilet and wearing clothes	21	2.5%
	During sleep	22	2.6%
	Without obvious reason	31	3.7%
	At all times	6	0.7%
	Other	22	2.6%

^1^ PCOS: polycystic ovary syndrome; ^2^ SUI: stress urinary incontinence.

**Table 3 medicina-59-00940-t003:** Association of risk factors with the presence of reported diagnosis of stress urinary incontinence among Saudi females investigated for stress urinary incontinence.

Variable	Categories	Presence of Reported Diagnosis of SUI		*p*-Value
Age groups (years)	20–29	4 (0.8%)		0.001 *
	30–39	9 (4.7%)		
	40–49	7 (6.3%)		
	50–59	8 (15.1%)		
	60 or more	0 (0%)		
Residence	Northern	2 (2.3%)		0.090
	Southern	1 (1.1%)		
	Central	8 (2.8%)		
	Eastern	2 (1.6%)		
	Western	15 (5.8%)		
Educational level	Primary	0 (0%)		0.699
	Intermediate	0 (0%)		
	Secondary	4 (2.6%)		
	Diploma	1 (1.6%)		
	Bachelor’s degree or above	23 (3.7%)		
Marital status	Single	2 (0.5%)		0.001 *
	Married	22 (5.9%)		
	Divorced	3 (7.7%)		
	Widowed	1 (16.7%)		
Average monthly income (SAR)	Less than 4000	3 (2.2%)		<0.001 *
	4000–8000	1 (1%)		
	More than 8000	19 (9.6%)		
	University incentive	1 (0.6%)		
	No monthly income	4 (1.8%)		
Body Mass Index (BMI) (kg/m^2^)	Underweight (<18.5)	0 (0)		<0.001 *
	Healthy weight (18.5–24.9)	6 (1.6%)		
	Overweight (25–29.9)	3 (1.3%)		
	Obese (>30)	19 (10.4%)		
Family history of SUI	Yes	13 (21%)		<0.001 *
	No	8 (1.5%)		
	I don’t know	7 (3%)		
History of Pregnancy	Yes	25 (7.1%)		< 0.001*
	No	3 (0.6%)		
History of smoking	Yes	2 (5.7%)		0.625
	No	26 (3.2%)		
		Presence of reported diagnosis of SUI		
		Yes	No	
		Mean (SD)		
Weight (Kg)		83 (18.1)	65 (16.7)	<0.001 *
Height (cm)		160 (21.6)	157 (15.4)	0.420

* *p* ≤ 0.05 is statistically significant, SUI: stress urinary incontinence, SAR: Saudi Riyal, kg/m^2^: kilogram per square meter, SD: standard deviation, kg: kilogram, cm: centimeter.

**Table 4 medicina-59-00940-t004:** Impact of SUI on participants’ everyday life.

Impact Score	Frequency (%)	Number of Cases Never Diagnosed with SUI ^1^ before
0	582 (69.1%)	581 (99.8%)
1	88 (10.5%)	88 (100%)
2	29 (3.4%)	29 (100%)
3	35 (4.2%)	35 (100%)
4	9 (1.1%)	8 (88.9%)
5	18 (2.1%)	14 (77.8%)
6	11 (1.3%)	9 (81.8%)
7	11 (1.3%)	9 (81.8%)
8	14 (1.7%)	11 (78.6%)
9	6 (0.7%)	5 (83.3%)
10	39 (4.6%)	25 (64.1%)
ICIQ-SF ^2^ score	Frequency (%)	Number of cases never diagnosed with SUI before
Slight (0–5)	689 (81.8%)	687 (99.7%)
Moderate (6–12)	112 (13.3%)	102 (91.1%)
Severe (13–18)	36 (4.3%)	24 (66.7%)
Very severe (19–21)	5 (0.6%)	1 (20%)

^1^ SUI: stress urinary incontinence; ^2^ ICIQ-SF: International Consultation on Incontinence Questionnaire Urinary Incontinence Short Form.

**Table 5 medicina-59-00940-t005:** Association between age, BMI, reported diagnosis of SUI, and ICIQ-SF score.

Variable	ICIQ-SF Score				*p*-Value
	Slight	Moderate	Severe	Very Severe	
	N (%)				
Age (years)					
20–29	421 (61.1)	49 (43.8)	10 (27.8)	0 (0)	<0.001 *
30–39	156 (22.6)	25 (22.3)	8 (22.2)	1 (20)	
40–49	71 (10.3)	28 (25)	12 (33.3)	1 (20)	
50–59	36 (5.2)	8 (7.1)	6 (8.3)	3 (60)	
60 or more	5 (0.7)	2 (1.8)	0 (0)	0 (0)	
Body mass index (BMI) (kg/m^2^)					
Underweight (<18.5)	49 (7.1)	9 (8)	0 (0)	0 (0)	< 0.001 *
Healthy weight (18.5–24.9)	319 (46.3)	33 (29.5)	11 (30.6)	1 (20)	
Overweight (25–29.9)	194 (28.2)	35 (31.3)	7 (19.4)	1 (20)	
Obese (>30)	127 (18.4)	35 (31.3)	18 (50)	3 (60)	
Presence of reported diagnosis of SUI					
Yes	2 (0.3)	10 (8.9)	12 (33.3)	4 (80)	<0.001 *
No	687 (99.7)	102 (91.1)	24 (66.7)	1 (20)	

* *p* ≤ 0.05 is statistically significant, ICIQ-SF: International Consultation on Incontinence Questionnaire Urinary Incontinence Short Form, SUI: stress urinary incontinence, kg/m^2^: kilogram per square meter.

**Table 6 medicina-59-00940-t006:** Predictors for SUI among Saudi females.

Variable	Odds Ratio	95% CI for Odds Ratio	*p*-Value
		(Lower–Upper)	
Age group (years)			0.582
20–29	1		
30–39	1.71	(0.34–8.60)	0.515
40–49	1.27	(0.19–8.64)	0.808
50–59	4.33	(0.52–36.32)	0.177
60 or more	0.00	(0.00)	0.999
Residence			0.241
Northern	1		
Southern	1.26	(0.05–31.69)	0.888
Central	3.97	(0.46–34.32)	0.211
Eastern	1.76	(0.13–23.82)	0.671
Western	6.70	(0.85–52.62)	0.070
Marital status			0.818
Single	1		
Married	2.49	(0.19–32.34)	0.485
Divorced	3.02	(0.19–48.51)	0.435
Widowed	7.15	(0.09–574.91)	0.380
Educational level			0.925
Primary	1		
Intermediate	1.11	(0.00)	1.000
Secondary	3.38	(0.00)	0.999
Diploma	8.21	(0.00)	0.999
Bachelor’s degree or above	1.93	(0.00)	0.999
Average monthly income (SAR)			0.164
Less than 4000	1		
4000–8000	0.40	(0.03–5.01)	0.474
More than 8000	4.03	(0.73–22.38)	0.111
University incentive	0.72	(0.05–11.15)	0.811
No monthly income	1.12	(0.18–7.17)	0.904
Family history of SUI			0.000
No	1		
Yes	19.68	(5.46–70.97)	0.000 *
I don’t know	4.35	(1.19–15.85)	0.026 *
Presence of history of pregnancy	2.58	(0.01–732.51)	0.742
Number of pregnancies			0.704
1	1		
2	0.49	(0.01–41.00)	0.755
3	0.14	(0.00–5.60)	0.296
4	0.35	(0.03–4.82)	0.434
5 or more	0.34	(0.04–3.17)	0.344
Number of vaginal deliveries			0.217
0	1		
1	4.59	(0.42–50.11)	0.212
2	3.63	(0.17–79.14)	0.413
3	5.96	(0.13–280.47)	0.364
4	0.62	(0.01–70.00)	0.845
5 or more	0.47	(0.00–101.46)	0.784
Number of caesarian sections			0.977
0	1		
1	1.12	(0.20–6.35)	0.903
2	2.32	(0.18–29.38)	0.517
3	1.60	(0.04–63.16)	0.802
4	4.94	(0.05–477.39)	0.494
5 or more	0.00	(0.00)	0.999
Smoking status	9.00	(0.99–82.29)	0.052

* *p* ≤ 0.05 is statistically significant, CI: confidence interval.

## Data Availability

The data presented in this study are available from the corresponding authors upon request.

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
