# Peer review of "Prevalence of Stress Urinary Incontinence and Risk Factors among Saudi Females"

_medicina, 2023, doi:10.3390/medicina59050940_

Round 1

Reviewer 1 Report

Dear Editor,

Thank you very much for giving me the opportunity to review this manuscript for your prestigious journal.

The present manuscript presents data from a descriptive cross-sectional study in a cohort of Saudi women with the aim of finding the prevalence by administration of the ICIQ-UI SF

Although the study has no novelty, because it is a widely studied topic, it does provide data from a less studied cohort, it is of great value to know the prevalence based on different ethnicities and even nationalities, and this cohort has not been widely studied. For that reason, I believe it may be of interest to Medicina readers.

This manuscript is well-written and it has an excellent scientific soundness.

However, before publication, the authors should review their manuscript, and I believe that the comments I make will strengthen the final document, and I invite you to take them into consideration.

GENERAL COMMENTS

I think that the justification of the study, in the part of the introduction, requires further development, in particular, if we talk about stress urinary incontinence, the practice of physical activity and/or sport should be contemplated, because it represents a very important risk factor for SUI.

Covariates that could influence the results and that, if not included, should be mentioned as limitations:

1-.Have the authors contemplated whether the women had received specific pelvic floor training?

2-. Had the women undergone any postpartum rehabilitation program?

The applied methodology is adequate for the research objective and the description of the processes is very complete.

The statistics are correctly applied.

The results are well explained, but I think they need further development in the discussion.

SPECIFIC COMMENTS

Line 18.  Background and objectives should appear. Please add it.

Line 36. The bibliographic reference of the IUGA is missing. Please add it.

In the introduction, it would be necessary to include physical activity and/or sport as a risk factor predisposing to SUI. I suggest add this topic.

Line 70. Pelvic surgical interventions (e.g. hysterectomy) or laparatomies have been considered. Please clarify

Line 82. ICIQ-UI SF criteria were applied for the diagnosis of SUI, i.e. more than 0 is the woman considered to have SUI? Please, explain

Line 97. How was the Odds Ratio calculated? Please, describe it.

Table. 1 should adjust the spacing to single line spacing.

The behavior of the ORs in relation to the number of normal deliveries is striking, and should be explained in the discussion.

Line 265. I do not understand this sentence, perhaps it should be rewritten.

What worries to me most (as the authors themselves comment) is the inequality of the sample segmented by age, the bulk of the age range is 20-29 years (57%) while 50-59 (6.3%) 60 or more (0.8%) I think that one way to solve this aspect is to segment the results by age range. Because when checking  the presence of SUI, women aged 60 and over have 0%, which is striking. The rest of the age range shows how the prevalence increases with age, with 50-59 years there is 15.1% therefore, I think it is very bold/daring to say that the prevalence in the female population is 3.3% because the differences in age are very large. I insist on segmenting the data by age groups, or at least creating two categories, young and adult.

And develop the results discussion and conclusions with age differences.

Finally, the references style should be revised and adjusted to the regulations of the journal.

Author Response

Thank you very much for giving me the opportunity to review this manuscript for your prestigious journal. The present manuscript presents data from a descriptive cross-sectional study in a cohort of Saudi women with the aim of finding the prevalence by administration of the ICIQ-UI SF. Although the study has no novelty, because it is a widely studied topic, it does provide data from a less studied cohort, it is of great value to know the prevalence based on different ethnicities and even nationalities, and this cohort has not been widely studied. For that reason, I believe it may be of interest to Medicina readers. This manuscript is well-written and it has an excellent scientific soundness. However, before publication, the authors should review their manuscript, and I believe that the comments I make will strengthen the final document, and I invite you to take them into consideration.

Response: Thank you for your constructive criticism and valuable comments.

GENERAL COMMENTS

I think that the justification of the study, in the part of the introduction, requires further development, in particular, if we talk about stress urinary incontinence, the practice of physical activity and/or sport should be contemplated, because it represents a very important risk factor for SUI.

Response: Thank you for your comment. We have addressed physical activity and/or sport as potential risk factors for SUI and cited a reference.

Covariates that could influence the results and that, if not included, should be mentioned as limitations:

1-Have the authors contemplated whether the women had received specific pelvic floor training?

2- Had the women undergone any postpartum rehabilitation program?

Response: Thank you for your comment. Unfortunately, these covariates were not examined. As suggested, we have acknowledged your point as a limitation in our limitations section. 

The applied methodology is adequate for the research objective and the description of the processes is very complete.

Response: Thank you for your positive remark.

The statistics are correctly applied.

Response: Thank you for your positive remark.

The results are well explained, but I think they need further development in the discussion.

Response: Response: Thank you for your positive remark. We have elaborated on more results in the discussion section and highlighted several limtations.

SPECIFIC COMMENTS

Line 18.  Background and objectives should appear. Please add it.

Response: We have added background and objectives headings separately.

Line 36. The bibliographic reference of the IUGA is missing. Please add it.

Response: We have added the citation.

In the introduction, it would be necessary to include physical activity and/or sport as a risk factor predisposing to SUI. I suggest add this topic.

Response: We have acknowledged your point and added a citation.

Line 70. Pelvic surgical interventions (e.g. hysterectomy) or laparatomies have been considered. Please clarify.

Response: We are not able to allocate the comment. Nevertheless, it is most probably a typo from our side and no further action is needed.

Line 82. ICIQ-UI SF criteria were applied for the diagnosis of SUI, i.e. more than 0 is the woman considered to have SUI? Please, explain

Response: Thank you for your comment. The ICIQ-UI SF was NOT used to diagnose SUI. The diagnosis of SUI is hinted subjectively by patient-reported signs/symptoms and then clinically diagnosed (established) via procedures, including physical examination, imaging, brief neurological exam, and specific tests (urinary stress test and 24-hour urine pad, etc). We relied on the subjective reporting of participants to establish diagnosis of SUI. The ICIQ-UO SF questionnaire was used to evaluate the frequency, severity and impact on quality of life of urinary incontinence in women. The scores of ICIQ-UO SF just reflect the severity level, as highlighted in Table 4: Slight (0 – 5), Moderate (6 – 12), Severe (13 – 18), and Very severe (19 – 21). In the limitations section, we highlighted these aspects.

Line 97. How was the Odds Ratio calculated? Please, describe it.

Response: The mathematical method of calculating odds ratio is well-documented in the published literature and referenced textbooks. However, we added a brief description in the methods section.

Table. 1 should adjust the spacing to single line spacing.

Response: we have adjusted the spacing for all tables, and the editorial office will take care of these matters during typesetting.

The behavior of the ORs in relation to the number of normal deliveries is striking, and should be explained in the discussion.

Response: we have added a brief explanation in the discussion section. Paragraphs 2 and 3 provide sufficient explanations, and our findings are line with the published literature.

Line 265. I do not understand this sentence, perhaps it should be rewritten.

Response: we have added a brief explanation in the discussion section.

What worries to me most (as the authors themselves comment) is the inequality of the sample segmented by age, the bulk of the age range is 20-29 years (57%) while 50-59 (6.3%) 60 or more (0.8%) I think that one way to solve this aspect is to segment the results by age range. Because when checking  the presence of SUI, women aged 60 and over have 0%, which is striking. The rest of the age range shows how the prevalence increases with age, with 50-59 years there is 15.1% therefore, I think it is very bold/daring to say that the prevalence in the female population is 3.3% because the differences in age are very large. I insist on segmenting the data by age groups, or at least creating two categories, young and adult.

And develop the results discussion and conclusions with age differences.

Response: Thank you for your comment. We agree with the inequality of sample size based on age. However, this is an unavoidable circumstance and represents a real-life experience, and the authors had no hand in it. Indeed, the prevalence of SUI increased with age, which is consistent with the published literature. We believe it is not necessary to modify the age groups into young and adults, and the presented range age groups should be kept as it is (In line with other studies). However, your valuable point about the limitation of age was clearly acknowledged in the limitations section.

Finally, the references style should be revised and adjusted to the regulations of the journal.

Response: we have revised the references style to match the journal’s style.

Reviewer 2 Report

Dear Authors,

Thank you for allowing me to review your manuscript.

The study is very interesting but to last the real not very original 

The introduction is well documented and the purpose of the study is clearly expressed.

I think it is necessary to carry out an in-depth analysis of the data on the subject present in the literature, perhaps with a table.

Finally, for a multidisciplinary approach we propose to mention:

DOI: 10.1186/s13027-022-00465-9 

DOI: 10.3390/biology11081114 

Author Response

Dear Authors. Thank you for allowing me to review your manuscript. The study is very interesting but to last the real not very original. The introduction is well documented and the purpose of the study is clearly expressed. I think it is necessary to carry out an in-depth analysis of the data on the subject present in the literature, perhaps with a table.

Response: Thank you for your comment. In consideration of the large number of tables associated with this manuscript, we opted to extensively discuss our findings textually in the discussion section.

Finally, for a multidisciplinary approach we propose to mention:

DOI: 10.1186/s13027-022-00465-9

DOI: 10.3390/biology11081114

Response: Thank you for your suggestion. We had a pleasure reading your two proposed citations. However, they are not directly related to our manuscript, and it is not appropriate to cite them. However, we will keep them in mind and cite them whenever needed in future manuscripts.

Reviewer 3 Report

Detecting and diagnose of SUI via online questionarre is suitable for epidemiologic or other informative research. collection of data from outpatients clinics where relevant data could be obtained would be more appropriate 

Author Response

Detecting and diagnose of SUI via online questionnaire is suitable for epidemiologic or other informative research. collection of data from outpatients clinics where relevant data could be obtained would be more appropriate.

Response: we have acknowledged your point as a limitation in our study.

Reviewer 4 Report

This is a well designed article. I feel the results can be described in a better and more simple way. 

Kindly recheck the language as the manuscript has minor spelling and punctuation issues. 

Author Response

This is a well-designed article. I feel the results can be described in a better and more simple way.

Response: Thank you for your suggestion. We have revised our results section and made it simpler.

Kindly recheck the language as the manuscript has minor spelling and punctuation issues.

Response: we have revised our manuscript for English and editorial comments.